# Network of Automated External Defibrillators in Poland before the SARS-CoV-2 Pandemic: An In-Depth Analysis

**DOI:** 10.3390/ijerph19159065

**Published:** 2022-07-25

**Authors:** Daniel Ślęzak, Marlena Robakowska, Przemysław Żuratyński, Kamil Krzyżanowski

**Affiliations:** 1Division of Medical Rescue, Faculty of Health Sciences with the Institute of Maritime and Tropical Medicine, Medical University of Gdańsk, 80-210 Gdańsk, Poland; przemyslaw.zuratynski@gumed.edu.pl (P.Ż.); kamil.krzyzanowski@gumed.edu.pl (K.K.); 2Department of Public Health & Social Medicine, Medical University of Gdańsk, 80-210 Gdańsk, Poland; marlena.robakowska@gumed.edu.pl; 3Department of Anesthesiology and Intensive Care, Oncology Center—Memorial Hospital in Bydgoszcz, 85-796 Bydgoszcz, Poland

**Keywords:** AED, automatic external defibrillator, AED in Poland

## Abstract

Introduction: Sudden cardiac arrest (SCA), which causes more than half of all cardiovascular related deaths, can be regarded as a common massive global public health problem. Analyzing out-of-hospital cardiac arrest (OHCA) cases, one of the key components is automatic external defibrillators (AEDs). Aim: The aim of this study was to analyze the use and distribution of AEDs in Polish public places. Materials and methods: The data were analyzed by using the Excel and R calculation programs. Results: The data represents 120 uses of automatic external defibrillators used in Polish public space in the period 2008–2018. The analysis describes 1165 locations of AEDs in Poland. It was noted that the number of uses in the period 2010–2016 fluctuated at a constant value, with a significant rise in 2017. When analyzing the time of interventions in detail the following was noted: the highest percentage of interventions was observed in April, and the lowest in November; the highest number of interventions was observed on a Friday, while the least number of interventions was observed on a Sunday; most occurred between 12:00 to 16:00, and least between 20:00 to 8:00. Conclusions: The observed growth in the number of cases of AED use in public places is associated with the approach to training, the emphasis on public access to defibrillation, and, therefore, the growth of social awareness. This study will be continued. The next analysis would include 2020–2022 and would be a comparative analysis with the current research.

## 1. Introduction

Defibrillation, as one of the links in the chain of survival, is an important part of the return of spontaneous circulation (ROSC). To perform this action, adequate equipment, namely, a defibrillator, is necessary. The general division of defibrillators distinguishes between manual devices (intended for medical personnel), and automatic and semi-automatic devices, which can also be used by laypeople. Regarding a semi-automatic device, the electrical shock is triggered by a witness to the event, and regarding an automatic device, the device carries out an analysis by itself as to whether defibrillation should be performed on a victim who has lost consciousness. AEDs follow current resuscitation guidelines and should be able to defibrillate between 120 and 360 J. Delgado et al. [1] recommend that rescuers who use a monophasic AED administer an initial shock with an energy of 360 J. This single dose for monophasic shocks is intended to simplify instructions for rescuers, but is not a mandate to withdraw monophasic AEDs for reprogramming. If the monophasic defibrillator AED in use is programmed to deliver a different first or subsequent dose, this dose is acceptable. The device should “guide the user by the hand”, step-by-step instructing the user what to do at any one time. The prompts provided by the AED should be underlined and understood by the potential user. It is emphasized that the prompts should interact with the CPR sequence to provide effectiveness (minimizing interruptions in chest compressions). An additional benefit is that the device has built-in sensors for precise chest compressions. It is good practice to equip the AED with extra accessories, such as shaving razors/pads, scissors, a mask for administering ventilation, or gloves [2,3]. Public Access to Defibrillation (PAD) programs are the result of proposed strategies to improve survival after SCA. The idea was first suggested by the AHA Task Force on Automated External Defibrillation in 1995. The first guidelines on AEDs were published by the European Resuscitation Council in 1998 [4]. In 2000, both societies published their guidelines on the use of AEDs and reinforced the need for PAD expansion. These were based on existing early defibrillation programs at airports and casinos. The recommendation was that AEDs should be placed where the risk of SCA is very high (at least one every two years) [5]. Meanwhile, the ERC and AHA have since amplified the idea of public access to defibrillation globally as one of the important components of the chain of survival [4,5,6,7,8].

The aim of this research was to investigate the current frequency and situation of the use of automated external defibrillators in public places in Poland, so the following research questions were asked: How often are AEDs used in public places? Are the number of AED uses affected by variables such as time of day, day of week, month and season? What is the profile of the place where AEDs were most often used?

## 2. Materials and Methods

The aim of the research was analysis of cases of the use of AEDs. Data were collected from a few resources: a questionnaire of a diagnostic interview survey (65 cases were compiled from this), answers from an author’s questionnaire sent to foundations promoting first aid and health (14 cases were compiled from this), and medical documentation (41 cases were compiled from this). After collecting the data, statistical analysis was conducted.

Research coverage consisted of information on the use of an automated external defibrillator in adults (over 18 years old) during the period reviewed (1 January 2008–31 December 2018). In total, 120 cases of use of an automated external defibrillator in a public place, other than a medical institute, were analyzed, while excluding the nation’s emergency services, i.e., fire departments and volunteer fire departments. These services have AED defibrillators, which are part of their emergency equipment.

The research material consisted of responses from a diagnostic survey questionnaire. The author’s questionnaire was sent to units with AEDs. Respondents’ answers to the questions recorded in the questionnaire were in accordance with their consent to participate in voluntary, cost-free research. The questionnaires were sent electronically. The questionnaire included the following questions:(1)Is/Are there an/any AED device(s) in the area you administer?(2)How long have you had an AED?(3)How many and what type of AED device do you have?(4)Is the AED located in an easily accessible place? Is it placed in a visible location?(5)Have any of the deployed AEDs been used in life-threatening situations for an employee or client?(6)Who has been using the AED?(7)Circumstances of the AED use. Please briefly describe.(8)Was the AED distributor/producer informed of the use of the device?(9)Has the AED distributor/manufacturer contacted the unit where the AED is located?

Another source of analyzed data were the responses from a questionnaire sent to foundations that promote and support the development of the principles of first aid and healthy lifestyles: the Polish Red Cross, the Great Orchestra of Christmas Charity Foundation, the World for Children Foundation, the Department of Bioinformatics and Telemedicine, Collegium Medium of the Jagiellonian University in Krakow, ORLEN “Gift of Health Foundation”. The questionnaire included questions:(1)Do you have any knowledge of the use of AEDs (2008–2018)?(2)In which circumstances have AEDs been used? Please give a brief description.

Additionally, there was a retrospective analysis of selected medical documentation. Data were obtained from emergency call center (medical dispatch center) reports, medical emergency cards, after approval from the dispatcher of the state emergency medical system and the management of emergency medical teams. The anonymized data included only the date and time of the event, age, gender and treatment administered. We analyzed data generated as a specific “type” of patient, from a specific disease entity (ICD-10 (I46)–SCA cardiac arrest) in a given period (time, day, month) for the years 2008–2018.

The research material was collected in a Microsoft Office Excel for Windows software spreadsheet. Statistical analyses were conducted using R 3.5.3 software (R version 3.5.3 (Great Truth, Gdansk, Poland) of 2019). Data were tested by Poisson tests with a significance value of α = 0.05.

## 3. Results

### 3.1. Frequency of Use of AEDs in Poland

The locations of AEDs were obtained from publicly available records and maps.

At the time of the survey, there were 1165 locations in Poland where an automated external defibrillator was installed (as of 1 January 2019, based on www.ratujzsercem.pl) [9]. This number refers to selected location sites, not to the total number of defibrillators, as many facilities have more than just one AED. Judging from the profiles of the places where the AEDs were located, it could be seen that the highest number of AEDs were found in commercial establishments, i.e., shopping malls, multi-branch stores (198). There was an equally high rate of occurrence in cultural establishments and institutions, i.e., museums, theaters, opera houses, art galleries, libraries (176). The lowest number were found in religious places, i.e., monasteries, churches, religious gathering places (13), and airports (12). The percentage and numbers are graphically presented in Figure 1.

### 3.2. The Use of AEDs by Responses to a Questionnaire Sent to Units with AEDs

The questionnaire was sent by e-mail to 1165 units with AEDs. The total of return responses was 326 (27.98%). Among all responses, reluctance to provide information on factual questions was noted, e.g., “For this reason, and due to our company’s procedures, unfortunately we cannot complete and send the questionnaire sent to you”, or “Due to the fact that the AED device was not used in our facility, we are unable to answer your question” (4.6% response rate).

Based on the questionnaire data, there was also a significant increase in the number of facilities that qualify for public access to AEDs between 2016 and 2018 (Table 1).

The overwhelming response from the responding units (78.52%) was that they had 1 AED. The place with the highest number of AEDs was Warsaw Chopin Airport. The airport has a system that consists of four components: 49 ready-to-use AEDs placed in the terminal and other airport facilities, 2 AEDs for emergency teams’ staff, as well as an automated emergency notification system and a training program for all airport employees. Reference should also be made to the response sent by Warsaw Metro. The institution has 35 AEDs. Regarding the location, each of the 31 stations, as well as the technical station at Kabaty, has a minimum of 1 AED. Therefore, each AED was assigned to a location (Table 2).

Some of the 63% of units interviewed responded as to what kind of company created the AED located in their location. Incomplete replies as to the device being an automatic device accounted for 37% of the respondents. Due to formal and legal considerations, specific models of AEDs are not presented, only a view of what technical parameters the most common device had. Semi-automatic versions dominated, using a biphasic defibrillation wave with a shock energy of 50 J for children and 50–150 J for adults.

A total of 54% of units ensured 24-h access to the AEDs. The other AEDs were only used during the relevant facility’s opening hours. Some AEDs were located at the customer/patient service points, in security rooms, etc. Only 20% of facilities were open 24 h a day and the AEDs were located in the public zone (outside the building).

According to responses from Warsaw Metro, AEDs were located at each subway station (in the middle of the platform) in a publicly accessible and signed area. In addition, a few AEDs were located in places not accessible to passengers, but accessible to Metro staff and services. Also noteworthy was the AED situation at Warsaw Chopin Airport, where, on average, 49 AEDs were located, one every 150 m.

### 3.3. Analysis of AED Use Cases

The analysis tested whether the number of AED interventions fluctuated from year to year, whether they fluctuated from month to month, from day to week, from season to season and from time of day (the following breakdown of the day was used: 0:00–4:00; 4:00–8:00; 8:00–12:00; 12:00–16:00; 16:00–20:00; 20:00–24:00). For each time unit, the number of interventions per year was converted, resulting in the number of interventions per year, along with a 95% confidence interval. The frequency of AED interventions by year, month, day of week, season and day were compared using the Poisson test.

Analysis of interventions by year revealed fluctuations between 2010 and 2016, but the rate was constant. There were no significant differences in the number of interventions by year. There was a larger significant increase in 2017 (*p* = 0.006). The frequency of interventions in 2018 was not significantly different from the previous year. The results are shown in Table 3 and Figure 2.

Analysis of the number of interventions by season showed no significant differences between seasons (*p* = 0.332). So, it turned out that the frequency of interventions did not differentiate between seasons. The results are shown in Table 4.

The monthly analysis of interventions showed that the highest number of interventions was in April and the lowest in November. There was a statistically significant difference between the abovementioned months. (*p* = 0.027). The frequency of interventions was not significantly different between the other pairs of months. The results are presented in Table 5.

Analyzing the number of interventions by day of the week, it could be seen that the number of interventions was highest on Fridays and lowest on Sundays. There was a statistically significant difference between the abovementioned days (*p* = 0.049) The frequency of interventions was not significantly different between the other pairs of days of the week. The results are presented in Table 6.

When analyzing the number of AED interventions for times of day, you will notice that the number of interventions was noticeably lower between 8.00 and 8.00 p.m. In this time period, the number of interventions did not differ significantly between different times of the day. Between 8:00 a.m. and 12:00 p.m., 12:00 p.m. and 4:00 p.m. and 8:00 p.m., the number of interventions was significantly higher than in the afore-mentioned times (*p* = 0.013; *p* < 0.001; *p* = 0.043, respectively). It is also worth noting that the number of interventions between 12:00 and 16:00 was significantly higher than those between 8:00 to 12:00 and between 16:00 to 20:00 (*p* = 0.015). The results are presented in Table 7.

## 4. Discussion

The aim of the study was to analyze the frequency of use of public AEDs in Poland, looking at the ten year period between 2008 and 2018, several years after the first AED installations in public places. Due to the low rate of feedback received from sites, we analyzed AED use in 120 cases. During the study period, 1165 sites were located where an automated external defibrillator was installed. This number only referred to specific places, not to the total number of AEDs in Poland, as many institutions had more than one AED. Most of the AEDs were bought and located in the respondent units in 2005 (as a result of the Great Orchestra of Christmas Charity Foundation event). It is also worth noting that in 2008–2009 there was an increase in the number of places where AEDs were located. This was due to the “Life Impulse” program implemented in the city of Krakow. Based on the methodology of the research, it could be concluded that the largest number of devices were located in shopping centers, i.e., shopping malls, multi-brand stores (17%). This group included store networks, i.e., Auchan, Jeronimo Martins Group “Biedronka”, Leroy Merlin, Tesco or Decathlon. The management of these store networks cares for the safety of their employees and customers. As presented by a spokesperson for the Biedronka brand, as part of the “Safe Biedronka” program, selected centers have been equipped with AED devices. The choice of locations was based on the high probability of using a defibrillator, due to the store’s distance from medical points and the presence of a large number of customers at the same time. In research by Slezak [9], who analyzed the number of AEDs in Polish voivodeship cities, shopping centers represented only 4%. The author analyzed the total number of AEDs, also taking into account medical and rescue service providers, i.e., fire, police and volunteer rescue groups, i.e., Polish Red Cross, Polish Scouting and Rescue Organization. By extrapolating the data Slezak received to the methodology adopted in the research, the percentage was 6.97%. According to the 2017 status presented by the authors of the “Rescue with Heart” initiative to map AEDs in Poland, in 2017 29.3% of the total number of AEDs were in stores and workplaces [10]. Due to the lack of other specific scientific reports about the location of AEDs in Poland, the research results could only be compared with those of Slezak [9] and the authors of “Rescue with Heart” [10]. Another group of locations namely, cultural and entertainment institutions, such as cinemas, theaters, opera houses, museums, art galleries and libraries, had an equally high rate of AED installation (15.1%). The “Multikino” cinema network, owned by Vue International Group, was the first to buy semi-automatic defibrillators for all 26 cinemas in Poland in 2012. Other cinema networks followed its example. AEDs are located in the Wieliczka Salt Mine near Krakow, the NOVA Opera House in Bydgoszcz, the St. Jaracz City Theater in Otwock, the Czestochowa Philharmonic, the Emigration Museum in Gdynia, among others. These are just some examples of places. The result was comparable to that reported by Slezak [9]. It was found that 11.42% of AEDs were located in state, government and territorial administration buildings, i.e., ministries, municipal/city/county offices. In Slezak’s research [9], these locations ranked first. Noteworthy were banking facilities (5.55%). One of the banking networks, ING Bank Slaski, has equipped its 38 bank offices with AEDs since 2013 and planned to purchase and install another 19 for 2020. All the devices were located in public places, so as to be available to anyone who needed them at any given time. A small number of AEDs were located at religious places, i.e., churches, monasteries, places of religious congregations (1.11%). Similar results were also reported by Slezak [9,11]. When presenting the AED deployment in other countries, we should mention Japan, where, in 2008, 25% of public AEDs were located in schools, 19% in workplaces or care institutions, 16% in workplaces, 4% in sports facilities, 3% in cultural facilities and 3% in public transportation. The distribution of AEDs was uncontrolled and depended on public and private initiatives [12,13]. In Hong Kong, by contrast, of the 1637 AEDs, 49.4% were deployed in educational facilities and 29.3% in recreational facilities (sports centers, parks, swimming pools, beaches, museums and libraries). Shopping centers accounted for only 4.5% [14]. The data collected by Moon et al. [15] showed that in Arizona (USA), the largest number of AEDs were located in places with the profile “public companies/offices/workplaces.” In an attempt to reach the aims of the research, the questionnaire was prepared with questions addressed to the units where AEDs were located. Of the total 1165 questionnaires, the total number of responses was 326 (27.98%). The responses were given in writing and by phone. In the case of 15 (4.6%) responses, negatives could be noted relating to giving information to factual questions, e.g., “Due to this, and our company’s procedures, unfortunately we cannot complete and send the survey” or “Due to the fact that the AED device was not used in our company, we are unable to answer your question”. There was no recognition that having an AED might show their facility in a good light, including being a place that cares about the safety of residents or employees. Due to the lack of legal regulations for reporting AED possession, they could not be forced to answer. When comparing the legislative aspects regarding reporting on automated external defibrillators in other countries, it is worth noting that Singapore has a detailed registry of AEDs. Establishments are obliged to record what the profile of their facility is, whether the AED is public or 24/7. The registry includes the device’s precise location [16]. Although countries acknowledge the important role that defibrillators have in OHCA, unfortunately there are no national regulations obliging their registration. In Japan, a registry has been attempted, but, due to the lack of regulations, this has not been possible [12,17]. Referring to the United States of America, it can be seen that the state regulations are different. As reported by the National Conference of State Legislatures, “(...) they generally address the availability of AEDs in public buildings, conditions of use, medical supervision, training requirements and post-event reports. Some states require schools to be equipped with AEDs, while others require their availability in health clubs or other fitness facilities.” [18,19].

The most common trend is to put one AED in each location (78.52%). Although there were some places that had more than one AED (21.48%). Such places were dominated by railroad infrastructure facilities, such as Bydgoszcz Central Station and Warsaw Central Station, airports, such as John Paul II International Airport Krakow-Balice and Warsaw Chopin Airport, and shopping malls, such as Galeria Morena in Gdansk and Golden Terraces in Warsaw. In sports and recreation centers, there was also an imperceptible increase in the number of AEDs. Noteworthy, was the attention to safety at subway stations in Warsaw, which, ss an institution, had 35 AEDs. In terms of location, each of the 31 stations and the Kabaty Technical Station had at least one device. AEDs in subway stations around the world are now standard. According to sources and our own observations, PADs are present in Munich (Germany), Osaka (Japan), Singapore (Singapore), Hong Kong (People’s Republic of China), Tokyo (Japan), Bangkok (Thailand), Chicago (USA), and Sao Paulo (Brazil). This is an understandable deployment procedure, due to ERC and AHA guidelines and OHCA statistics [20,21,22].

AEDs should be located in publicly accessible areas where potential witnesses to an incident have 24-h access. According to survey respondents, 54% of facilities allowed 24-h access to AEDs. The remaining AEDs were only available during the units’ business hours. Only 20% of the 24/7 locations were outside the building (e.g., the building facade). If needed, the device should be usable immediately. As a 2008 journalistic report showed, in Krakow, after the start of Krakow’s IMPULSE OF LIFE AED program, AED dispensing was very problematic. In spite of staff training and a media campaign in places where defibrillators were available, witnesses to an incident had significant problems obtaining the device, which is important from the point of view of the “chain of survival” [23]. From the information obtained, it appeared that the devices were placed in buildings, e.g., at customer/patient service points, security rooms. For the system to function properly, education of those closest to, or within reach of, the AED should be an important element. The collected data suggested that between 2008 and 2018 there were 120 cases of AED use in a public place. This result could have been higher, but the design of the research methodology and the general resistance of the respondent institutions resulted in the figure given. Comparing these data with those from other countries or cities, the results were not impressive. In a one-year observational study (06.2017–06.2018) conducted by Alqahtani et al. [24] in the United Arab Emirates, there were only 13 cases of use of public defibrillators (1.8%) out of 715 OHCA cases. According to a study by Kitamura et al. [12] conducted between 2005 and 2007 in Japan, the ratio of AED use to total OHCA was 3.65% (*n* = 462). In a study conducted by Ringh [25] between 2006 and 2012 in Stockholm, Sweden, 6532 cases of OHCA were recorded. Of these, 474 occurred in public places where AED access was provided. Only 74 cases (16%) involved the use of an automatic defibrillator.

The number of uses of automatic defibrillators in Poland is rising every year. Between 2010 and 2016, the number oscillated at a constant level. A significant and statistically significant increase was observed in 2017 (*n* = 34; 28%; *p* = 0.0060). A non-statistical increase also occurred in 2018 (*n* = 40; 33%; *p* > 0.05). It can be assumed that this is due to an increase in public awareness of CPR efforts and an increasing number of AED defibrillators. However, there is no scientific reference, as no studies have been conducted.

Representing AED use in terms of a unit of time, there is no support for the hypothesis that season affects defibrillator resuscitation interventions. This indicates that climatic factors do not affect OHCA and AED use. However, there are noticeable differences when individual months are analyzed. The highest number of OHCAs in which the witness used an AED was in April (*n* = 16; 13.33%), and the lowest in November (*n* = 5; 4.16%). The difference between the mentioned months is statistically significant (*p* = 0.027). At the limit of statistical significance (*p* = 0.049), it was shown that AEDs were used more often on Fridays (*n* = 22; 18.33%) than on Sundays (*n* = 10; 8.33%). Time of day may also influence the number of AEDs used. It was noted that the highest number of OHCAs was between 12:00 and 16:00 (*n* = 39; 32.5%; *p* < 0.001). Translating this into a day/night split, i.e., from 8:00–20:00/20:00–8:00, 70% of cases occurred during the day and 30% at night. There are no detailed statistics and studies in the Polish and world literature to explain the above relationships and the results obtained. It can be concluded that this has an impact on the availability of AEDs. The lack of 24-hour access to AEDs on Sundays and at night is due to the fact that a significant number of them are located in buildings where no one stays on weekends and at night. There is also a shortage of potential patients who may suffer cardiac arrest, such as those in the workplace. Hansen et al. [26] noted that in the Copenhagen area (Denmark), most AEDs were available during the day every day of the week. Only 9.1% (*n* = 50) of all AEDs were available 24 hours a day, 7 days a week. To achieve the expected results, i.e., a reduction in OHCA mortality, continuous access to defibrillation should be pursued. The results obtained can only be translated into general studies on OHCA. In a study conducted by Bagai et al. [27] between 1 October 2005 and 31 December 2010, significant variation was observed in the incidence of OHCA by time of day (*p* < 0.001), day of week (*p* < 0.001) and month of year (*p* < 0.001), with the highest incidence occurring from Friday to Monday in December. Similar findings were made by Nadolny et al. [28], noting that OHCA occurred most frequently in the first quarter of the year, between 07:00 and 19:00. Szczerbinski et al. [29] observed seasonal differences in the incidence of OHCA, which may be influenced by temperature. The author reported that this should be subjected to further research analysis. This research will continue. The next analysis will cover the years 2020–2022 and will be a comparative analysis to the current analysis.

## 5. Conclusions

Since 2016, an increase in the frequency of use of AED devices, accessible in public places in Poland, can be observed (an increase in use of about 35%). This may be the result of the spread of public access to defibrillation, promotional campaigns and increased social self-awareness. The use of AEDs does not depend on the season, but there is a strong correlation by month (most in April, least in November), by day of the week (most on Friday, least on Sunday), and by time of day (most between 8 a.m. and 4 p.m., least between 8 a.m. and 4 a.m.). This research will continue. The next analysis will cover 2020–2022 and will be a comparative analysis to the current research.

## Figures and Tables

**Figure 1 ijerph-19-09065-f001:**
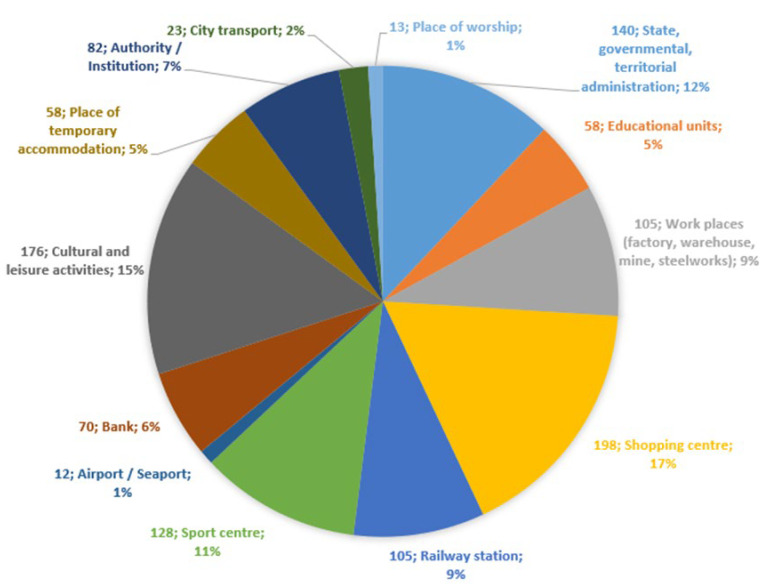
Profile of the location of AED devices (*n* = 1165).

**Figure 2 ijerph-19-09065-f002:**
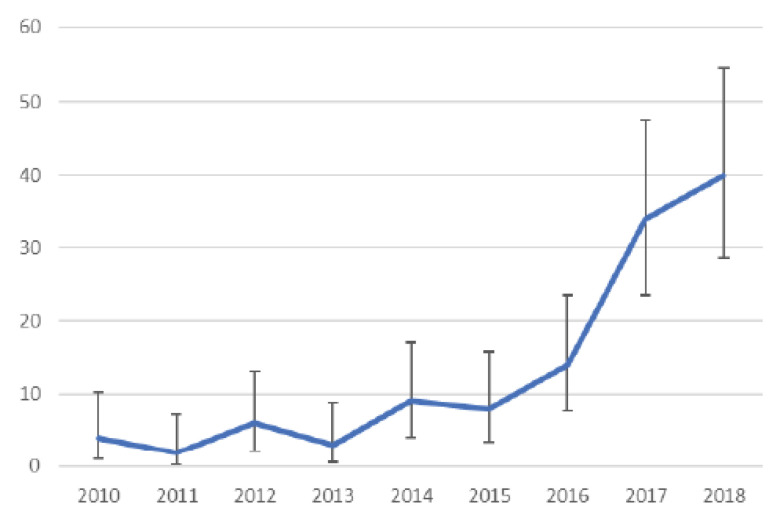
Comparison of individual years in terms of frequency of AED interventions.

**Table 1 ijerph-19-09065-t001:** The year of installation of the AED at the location.

Year of Placement of the AED	*n*	%
2004	2	0.67
2005	49	16.44
2006	7	2.35
2007	6	2.01
2008	33	11.07
2009	24	8.05
2010	13	4.36
2011	7	2.35
2012	10	3.36
2013	9	3.02
2014	11	3.69
2015	10	3.36
2016	41	13.76
2017	45	15.10
2018	31	10.40

**Table 2 ijerph-19-09065-t002:** A number of AED devices per unit (*n* = 298).

Number of AEDs per Location	*n*	%
1 AED	234	78.52
2–10 AEDs	54	18.12
above 10 AEDs	10	3.36

**Table 3 ijerph-19-09065-t003:** Frequency of AED interventions by year.

Year	Number of AED Interventions	95% CI ^1^	
		LL	UL
2010	4	1.09	10.24
2011	2	0.24	7.22
2012	6	2.20	13.06
2013	3	0.62	8.77
2014	9	4.12	17.08
2015	8	3.45	15.76
2016	14	7.65	23.49
2017	34	23.55	47.51
2018	40	28.58	54.47

^1^ 95% CI—confidence interval; LL and UL—lower and upper bounds of the confidence interval.

**Table 4 ijerph-19-09065-t004:** Frequency of AED interventions by season.

Season	AED Intervention/Year Rate	95% CI ^1^	
		LL	UL
spring	11.6	7.77	16.66
summer	8.8	5.51	13.32
autumn	10.4	6.79	15.24
winter	12.0	8.10	17.13

^1^ 95% CI—confidence interval; LL and UL—lower and upper bounds of the confidence interval.

**Table 5 ijerph-19-09065-t005:** Frequency of AED interventions by month.

Month	AED Intervention/Year Rate	95% CI ^1^	
		LL	UL
January	13.2	6.59	23.62
February	12.0	5.75	22.07
March	9.6	4.14	18.92
April	19.2	10.97	31.18
May	8.4	3.38	17.31
June	8.4	3.38	17.31
July	7.2	2.64	15.67
August	8.4	3.38	17.31
September	14.4	7.44	25.15
October	9.6	4.14	18.92
November	6.0	1.95	14.0
December	12.0	5.75	22.07

^1^ 95% CI—confidence interval; LL and UL—lower and upper bounds of the confidence interval.

**Table 6 ijerph-19-09065-t006:** Frequency of AED interventions by day of the week.

Day of Week	AED Intervention/Year Rate	95% CI ^1^	
		LL	UL
Monday	11.2	6.40	18.19
Tuesday	11.2	6.40	18.19
Wednesday	8.4	4.34	14.67
Thursday	11.2	6.40	18.19
Friday	15.4	9.65	23.32
Saturday	10.5	5.88	17.32
Sunday	7.0	3.36	12.87

^1^ 95% CI—confidence interval; LL and UL—lower and upper bounds of the confidence interval.

**Table 7 ijerph-19-09065-t007:** Frequency of AED interventions by the time of day.

Time of Day	AED Intervention/Year Rate	95% CI ^1^	
		LL	UL
0:00–4:00 a.m.	1.2	0.15	4.33
4:00–8:00 a.m.	4.8	2.07	9.46
8:00–12:00 a.m.	14.4	9.23	21.43
12:00–4:00 p.m.	23.4	16.64	31.99
4:00–8:00 p.m.	12.6	7.8	19.26
8:00–00:00 p.m.	5.4	2.47	10.25

^1^ 95% CI—confidence interval; LL and UL—lower and upper bounds of the confidence interval.

## Data Availability

Data sharing not applicable.

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
