# Peer review of "Network of Automated External Defibrillators in Poland before the SARS-CoV-2 Pandemic: An In-Depth Analysis"

_ijerph, 2022, doi:10.3390/ijerph19159065_

Round 1
Reviewer 1 Report
1. Please provide IRB approval number.
2. Please check your reference number. It is not organized.
3. Introduction part should be restructured. The contents are disconnected. And I don't understand what you are saying.
4. In the discussion, you should explain what your results meant.
5. Conclusion might be abrupt.
"This can be associated with the dissemination of public access to defibrillation, promotional campaigns and an increase in social awareness."
For the above sentence, you would better find and add the research about the association between the dissemination of public access to defibrillation, promotional campaigns, and an increase in social awareness.
Author Response
Reviewer 1 (Round 1)
Thank you very much for your objective review. We will refer to the comments below.
- Please provide the IRB approval number.
Answer 1: Information to the IRB is at the end of the manuscript: Institutional Review Board Statement: Patient consent was waived due to REASON (the data were obtained from the analysis of medical records, and no data can be traced to individual patients).
- Please check your reference number. It is not organized.
Answer 2: Thank you very much for your valuable comment. We have reorganized the references at the end of the manuscript and the references in the text.
- Delgado H, Toquero J, Mitroi C, Castro V, Fernández Lozano I. Principles of External Defibrillators. Available online: https://www.intechopen.com/chapters/41776 (accessed on 11.09.2021).
- Adult basic resuscitation and automated external defibrillation. Available online: [W:] https://www.prc.krakow.pl/wyt2015/2_BLS.pdf (accessed 30.09.2021).
- Trybus-Gałuszka H, Sokołowska-Kozub T. Defibrillation. Available online: [W:] https://www.prc.krakow.pl/wyd/2006/skrypt_2006-AED.pdf (accessed 30.09.2021)
- Part 1: Introduction to the International Guidelines 2000 for CPR and ECC. A Consensus on Science. Available online: https://www.ahajournals.org/doi/10.1161/circ.102.suppl_1.I-1 (accessed 30.01.2020)
- Skonieczny G, Marciniak M, Jaworska K. Sudden cardiac arrest - possibilities of defibrillation in primary and secondary prevention Forum Med. Rodz. 2012, 6(6), pp.:283–290.
- Early Defibrillation Task Force of the European Resuscitation Council. The 1998 European Resuscitation Council guidelines for the use of automated external defibrillators by EMS providers and first responders. Resuscitation 1998, 37, pp.:91–94.
- The Public Access Defibrillation Trial Investigators. Public-access defibrillation and survival after out-of-hospital cardiac arrest. N Engl J Med. 2004, 351:, pp.637–646.
- Priori SG, Bossaert LL, Chamberlain DA, Napolitano C, Arntz HR, Koster RW, et al. ESC-ERC recommendations for the use of automated external defibrillators (AEDs) in Europe. Euro Heart J. 2004, 25(5), pp.:437-445.
- Ślęzak D. Analysis of the availability of automated external defibrillators in Polish voivodship cities. Doctoral dissertation. Medical University of Karol Marcinkowski in Poznań. 2014. [w:] http://www.wbc.poznan.pl/Content/328208/PDF/index.pdf [dostęp on-line z dnia 10.02.2020 r.].
- „Ratuj z sercem”. AED localization. Available online: http://www.ratujzsercem.pl/Onas/Aktualno%C5%9Bci.aspx?news=Lokalizacje-AED (accessed on 10.02.2020 r.)
- AED defibrillator in the church AED. Available online: https://projektaed.pl/fundacja/aed-w-kosciele/ (accessed on 10.02.2020)
- Kitamura T, Iwami T, Kawamura T, Nagao K, Tanaka H, Hiraide A; Implementation Working Group for the All-Japan Utstein Registry of the Fire and Disaster Management Agency. Nationwide public-access defibrillation in Japan. N Engl J Med. 2010, 362(11), pp.:994-1004.
- Mitamura H. Public access defibrillation: advances from Japan. Nat Rev Cardiol 2008;, 5:, pp. 690–692.
- Fan M, Fan KL, Leung LP. Walking Route–Based Calculation is Recommended for Optimizing Deployment of Publicly Accessible Defibrillators in Urban Cities. Am Heart Assoc. 2020, ;9, :e014398. DOI: 10.1161/JAHA.119.014398.
- Moon S, Vadeboncoeur TF, Kortuem W, Kisakye M, Karamooz M, White B, Brazil P, Spaite DW, Bobrow BJ. Analysis of out-of-hospital cardiac arrest location and public access defibrillator placement in Metropolitan Phoenix, Arizona. Resuscitation. 2015;, 89, pp. :43-49.
- List of verified public access AED Locations. Government of Singapore. Available online: https://data.gov.sg/dataset/list-of-verified-public-access-aed-locations [dostęp online z DNA(accessed on 30.01.2020) r.].
- Kiyohara K, Kitamura T, Sakai T, Nishiyama C, Nishiuchi T, Hayashi Y, Sakamoto T, Marukawa S, Iwami T. Public- access AED pad application and outcomes for out-of-hospital cardiac arrests in Osaka, Japan. Resuscitation. 2016, 106, pp.:70-75.
- State Laws on Cardiac Arrest and Defibrillators. NCSL. Available online: https://www.ncsl.org/research/health/laws-on-cardiac-arrest-and-defibrillators-aeds.aspx (accessed on 30.01.2020) r.].
- An Overview of State AED Laws and Recommendations. Available online: https://www.cardiopartners.com/blog/an-overview-of-state-aed-laws-and-recommendations (accessed on 10.02.2020 r.]).
- Cardiac Science. Seven-year AED study in Munich: Cardiac Science for the week of May 18. Available online. https://www.cardiacscience.com/seven-year-aed-study-in-munich-cardiac-science-for-the-week-of-may-18/ (accessed on: 10.02.2020 r).
- Installation of AED (Automated External Defibrillator) units. Osaka Metro. Available online: https://subway.osakametro.co.jp/en/guide/usage/aed/aed_0.php (accessed on: 10.02.2020 r)
- Gianotto‐Oliveira R, Gonzalez MM, Vianna CB, Alves MM, Timerman S, Filho RK, Karl B. Survival After Ventricular Fibrillation Cardiac Arrest in the Sao Paulo Metropolitan Subway System: First Successful Targeted Automated External Defibrillator (AED) Program in Latin America. JAHA 2015, ;4(10).
- TOK FM. Hidden camera: "I can't give out the defibrillator!" Łukasz Wojtusik, TOK FM. Available online: https://wiadomosci.gazeta.pl/wiadomosci/1,114873,6265862,Ukryta_kamera___Nie_moge_wydac_defibrylatora__.html (accessed on: 10.02.2020 r)
- Alqahtani SE, Alhajeri AS, Ahmed AA, Mashal SY. Characteristics of Out of Hospital Cardiac Arrest in the United Arab Emirates. Heart Views. 2019, ;20(4):, pp.146-151.
- Ringh M, Jonsson M, Nordberg P, Fredman D, Hasselqvist-Ax I, Håkansson F, Claesson A, Riva G, Hollenberg J. Survival after Public Access Defibrillation in Stockholm, Sweden--A striking success. Resuscitation. 2015, ;91, pp.:1-7.
- Hansen CM, Wissenberg M, Weeke P, Ruwald MH, Lamberts M, Lippert FK, Gislason GH, Nielsen SL, Købe L, Torp-Pedersen C, Folke F. Automated External Defibrillators Inaccessible to More Than Half of Nearby Cardiac Arrests in Public Locations During Evening, Nighttime, and Weekends. Circulation 2013;, 128(20), pp.: 2224-2231.
- Bagai A, McNally B, Al-Khatib S, et al. Temporal differences in out-of-hospital cardiac arrest incidence and survival. Circulation, 2013, ;128, pp.:2595-2602.
- Nadolny K, Gotlib J, Panczyk M, Ładny JR, Białczak Z, Podgórski M, Makar O, Izhytska N, Gałązkowski R. Epidemiology of sudden cardiac arrest in prehospital care in the Silesian Wiadomości Lekarskie 2018, ;71(1), pp.:193-200.
- Szczerbinski S, Ratajczak J, Lach P, et al. Epidemiology and chronobiology of out-of-hospital cardiac arrest in a subpopulation of southern Poland: A two-year observation. Cardiol J. 2018 [Epub ahead of print]
- Introduction part should be restructured. The contents are disconnected. And I don't understand what you are saying.
Answer 3: Thank you very much for your valuable comment. The introduction is structured to first describe the AED device itself and then its importance in SCA. In addition, there is the purpose of the work.
- In the discussion, you should explain what your results meant.
Answer 4: Thank you very much for your valuable comment. The discussion provides an explanation of the results, for example:
The number of uses of automated defibrillators in Poland is increasing every year. Over the period 2010-2016, this number fluctuated at a constant level. A significant and statistically significant increase was observed in 2017 (n=34; 28%; p = 0.0060). A non-statistically significant increase also occurred in 2018 (n=40; 33%; p > 0.05). It can be presumed that this is due to increasing public awareness of resuscitation activities and the increasing number of AEDs. However, there is no scientific reference as no studies have been conducted.
Despite staff trainings and media campaign in the places where defibrillators were available, post-operative users encountered significant problems with receiving the device, im-portant from the point of view of the "survival chain" [22].[23]. From the information ob-tained, the devices were placed in buildings, e.g. at customer/petient service points, security rooms. For proper functioning of the system, an important element should be the education of the persons closest to the AED or within its range.
- Conclusion might be abrupt.
Answer 5: Thank you very much for your valuable comment. In the conclusion, we have reposted what we consider to be the most important information resulting from the study.
"This can be associated with the dissemination of public access to defibrillation, promotional campaigns and an increase in social awareness."
For the above sentence, you would better find and add the research about the association between the dissemination of public access to defibrillation, promotional campaigns, and an increase in social awareness.
We do not find confirmation of this condition in the available studies.

Reviewer 2 Report
I would kindly like to thank you for the opportunity to review the article above. Improvements in the field of resuscitation are necessary because of high mortality attributed to OHCA, which has been outlined in the article. However, I have a number of comments regarding said article.
Extensive english language editing is required throughout the manuscript.
The structure of the manuscript needs to be improved - the introduction section is too extensive and needs to be shortened.
In the methods section it is not entirely clear how the data was obtained - questionnaire or retrospective data? Why use questionnaire if retrospective data is available? Is location data available? Why not compare locations of cardiac arrests with locations of AEDs? If questionnaires are used, why use open, unstructured questions?
The results section needs to be rearranged according to revised methods section.
The discussion section needs to be rearranged according to revised methods and results sections. Comparison of your data with other similar articles needs to emphasised, with consideration of different economic, social and cultural circumstances. Currently the emphasis in the discussion section is on the description of the current status in Poland - other media are suitable for that role.
Author Response
Reviewer 2 (Round 1)
Thank you very much for your objective review. We will refer to the comments below.
- Extensive english language editing is required throughout the manuscript.
Answer 1: Thank you very much for your valuable comment. There will be another editing for linguistic correctness.
- The structure of the manuscript needs to be improved - the introduction section is too extensive and needs to be shortened.
Answer 2: Thank you very much for your valuable comment. The introduction contains only the most important information on the topic presented, that is, a brief discussion of the defibrillator and its importance in resuscitation.
- In the methods section it is not entirely clear how the data was obtained - questionnaire or retrospective data? Why use questionnaire if retrospective data is available? Is location data available? Why not compare locations of cardiac arrests with locations of AEDs? If questionnaires are used, why use open, unstructured questions?
Answer 3: Thank you very much for your valuable comment. We structured the study's methodologies as follows:
step 1. creation of a list of AEDs in Poland.
step 2. creating a questionnaire of questions (open-ended questions to obtain as much information as possible about the location of the AED and a possible episode of AED use).
Step 3. sending questionnaires to places with AEDs and AED distributors in Poland and public health foundations.
step 4. obtaining retrospective data from medical records.
The methodology of the study was to contribute to obtaining more data and therefore a reliable evaluation of the actual state of affairs.
The methodology was sufficiently described, including the presentation of the questions from the questionnaire.
Data on the location of the AED is available from the authors of the study.
The study was based on an analysis of AED use in SCA, not an analysis of SCA versus AED availability.
- The results section needs to be rearranged according to revised methods section.
Answer 4: Thank you very much for your valuable comment. In the Results section, we present 3 subsections:
3.1. Frequency of AED use in Poland
3.2. Use of AEDs according to the answers to the questionnaire sent to the units
3.3. Analysis of AED use cases
The first section presents AED locations (sites and their profiles) from publicly available maps of AEDs in Poland. The second chapter presents the use of AEDs in a given location from data obtained from the questionnaire. The third chapter presents use cases and their detailed course. The results in the third chapter are data from the questionnaire of questions and medical records.
- The discussion section needs to be rearranged according to revised methods and results sections. Comparison of your data with other similar articles needs to emphasised, with consideration of different economic, social and cultural circumstances. Currently the emphasis in the discussion section is on the description of the current status in Poland - other media are suitable for that role.
Answer 5: Thank you very much for your valuable comment. In the discussion, we also present a comparison with other countries and research teams, for example:
" In contrast, in Hong Kong, out of 1637 AEDs, 49.4% were installed in educational facilities and 29.3% in recreational facilities (sports centers, parks, swimming pools, beaches, museums, and libraries). Shopping centers accounted for only 4.5% [14]. From the data compiled by Moon et al [15], within Arizona (USA), the largest number of AEDs were located in places with a "public company/office/workplace" profile."
"Although states recognize the critical role that defibrillators play in OHCA, unfortunately there are no national regulations requiring providers to register them. In Japan, an attempt was made to establish a registry, but due to the lack of legislation, this was not possible [12,17]. Recalling the United States of America, it can be noted that the state laws are different. As presented by the National Conference of State Legislatures "(...) they generally address the availability of AEDs in public buildings, conditions of use, medical supervision, training requirements and postevent reports. Some states require schools to be equipped with AEDs, while others require their availability in health clubs or other fitness facilities."[18,19].".
"AEDs in subway stations around the world are already a standard. According to source data and own observations, PADs are present in Munich (Germany), Osaka (Japan), Singa-pore (Singapore), Hong Kong (People's Republic of China), Tokyo (Japan), Bangkok (Thailand), Chicago (USA), Sao Paulo (Brazil). This is an understandable placement procedure due to ERC and AHA guidelines and OHCA statistics [20-22]."

Reviewer 3 Report
Dear Sir/Madam,
I had the opportunity to act as a reviewer on the recent submission by Ślęzak et al. to the International Journal of Environmental Research and Public Health.
in this manuscript Ślęzak et al. present a very interesting study investigating the use and distribution of external defibrillators in the Polish public space before the pandemic. The authors found that the highest number of interventions took place on Friday, while the fewest interventions were recorded on Sunday the highest from 12:00 to 16:00, and the lowest from 20:00 to 8:00.
The article brings new knowledge in structure and organisation of the AED network in Poland.
However, some issues need to be addressed:
- I recommend introducing the typical structure (Background, aim etc.) in the abstract.
- I recommend proofing of English language by native speaker as well as eliminating the unnecessary hyphens (“-“) throughout the abstract and manuscript text.
- I fail to understand what criterion was chosen for ordering the references in text (neither in the order of first appearance nor in alphabetically): I recommend choosing a strategy from the both, it can become more user-friendly.
- Please transform “Anonymized data” (line 103) in a proper sentence.
- I recommend explaining the prepandemic context not only in the title but also in the abstract and manuscript.
- The article title is somewhat vague: I would reformulate it as follows: “Network of automated external defibrillators in Poland before the SARS-CoV-2 pandemic: an in-depth analysis”
Best regards
Author Response
Reviewer 3 (Round 1)
Thank you very much for your objective review. We will refer to the comments below.
- I recommend introducing the typical structure (Background, aim etc.) in the abstract.
Answer 1: Thank you very much for your valuable comment. We added the words the reviewer asked for.
Abstract: Introduction: Sudden cardiac arrest (SCA), which causes more than half of all deaths from cardiovascular causes, can be considered a widespread mass problem of public health around the world. When analyzing out-of-hospital cardiac arrest (OHCA), one of the key components is automated external defibrillators (AEDs). Aim: The research goal was to analyze the use and distribution of external defibrillators in the Polish public space. Materials and Methods: The data was analyzed thanks to the work with the Excel calculation program and the R program. Results: The data shows 120 applications of the automatic external defibrillator used in the Polish public space in the years 2008-2018. The analysis describes 1,165 AED installation sites in Poland. It was noted that the number of uses in 2010-2016 fluctuated, within a constant level, and there was a significant increase in 2017. Analyzing in detail the time of intervention:; the highest percentage of interventions was observed in April and the lowest in November; while the highest number of interventions was observed on Friday, while the fewest interventions were recorded on Sunday the highest from 12:00 to 16:00, and the lowest from 20:00 to 8:00. Conclusions: The observed development of the number of AED use cases in the public space is related to the approach to training - emphasis on the universal access to public access to defibrillation, and thus the development of social awareness.
- I recommend proofing of English language by native speaker as well as eliminating the unnecessary hyphens (“-“) throughout the abstract and manuscript text.
Answer 2: Thank you very much for your valuable comment. We will make a linguistic analysis of the text. Removed unnecessary hyphens (-).
- I fail to understand what criterion was chosen for ordering the references in text (neither in the order of first appearance nor in alphabetically): I recommend choosing a strategy from the both, it can become more user-friendly.
Answer 3: Thank you very much for your valuable comment. We have corrected the references according to the citation order. We have reorganized the references at the end of the manuscript and the references in the text.
- Delgado H, Toquero J, Mitroi C, Castro V, Fernández Lozano I. Principles of External Defibrillators. Available online: https://www.intechopen.com/chapters/41776 (accessed on 11.09.2021).
- Adult basic resuscitation and automated external defibrillation. Available online: [W:] https://www.prc.krakow.pl/wyt2015/2_BLS.pdf (accessed 30.09.2021).
- Trybus-Gałuszka H, Sokołowska-Kozub T. Defibrillation. Available online: [W:] https://www.prc.krakow.pl/wyd/2006/skrypt_2006-AED.pdf (accessed 30.09.2021)
- Part 1: Introduction to the International Guidelines 2000 for CPR and ECC. A Consensus on Science. Available online: https://www.ahajournals.org/doi/10.1161/circ.102.suppl_1.I-1 (accessed 30.01.2020)
- Skonieczny G, Marciniak M, Jaworska K. Sudden cardiac arrest - possibilities of defibrillation in primary and secondary prevention Forum Med. Rodz. 2012,; 6(6), pp.:283–290.
- Early Defibrillation Task Force of the European Resuscitation Council. The 1998 European Resuscitation Council guidelines for the use of automated external defibrillators by EMS providers and first responders. Resuscitation 1998, ;37, pp.:91–94.
- The Public Access Defibrillation Trial Investigators. Publicaccess defibrillation and survival after out-of-hospital cardiac arrest. N Engl J Med. 2004, ;351:, pp.637–646.
- Priori SG, Bossaert LL, Chamberlain DA, Napolitano C, Arntz HR, Koster RW et al. ESC-ERC recommendations for the use of automated external defibrillators (AEDs) in Europe. Euro Heart J. 2004, ;25(5), pp.:437-445.
- Ślęzak D. Analysis of the availability of automated external defibrillators in Polish voivodship cities. Doctoral disserta-tion. Medical University of Karol Marcinkowski in Poznań. 2014. [w:] http://www.wbc.poznan.pl/Content/328208/PDF/index.pdf [dostęp on-line z dnia 10.02.2020 r.].
- „Ratuj z sercem”. AED localization. Available online: http://www.ratujzsercem.pl/Onas/Aktualno%C5%9Bci.aspx?news=Lokalizacje-AED (accessed on 10.02.2020 r.)
- AED defibrillator in the church AED. Available online: https://projektaed.pl/fundacja/aed-w-kosciele/ (accessed on 10.02.2020)
- Kitamura T, Iwami T, Kawamura T, Nagao K, Tanaka H, Hiraide A; Implementation Working Group for the All-Japan Utstein Registry of the Fire and Disaster Management Agency. Nationwide public-access defibrillation in Japan. N Engl J Med. 2010, ;362(11), pp.:994-1004.
- Mitamura H. Public access defibrillation: advances from Japan. Nat Rev Cardiol 2008;, 5:, pp. 690–692.
- Fan M, Fan KL, Leung LP. Walking Route–Based Calculation is Recommended for Optimizing Deployment of Publicly Accessible Defibrillators in Urban Cities. Am Heart Assoc. 2020, ;9, :e014398. DOI: 10.1161/JAHA.119.014398.
- Moon S, Vadeboncoeur TF, Kortuem W, Kisakye M, Karamooz M, White B, Brazil P, Spaite DW, Bobrow BJ. Analysis of out-of-hospital cardiac arrest location and public access defibrillator placement in Metropolitan Phoenix, Arizona. Resus-ci-tation. 2015;, 89, pp. :43-49.
- List of verified public access AED Locations. Government of Singapore. Available online: https://data.gov.sg/dataset/list-of-verified-public-access-aed-locations [dostęp on-line z dnia(accessed on 30.01.2020) r.].
- Kiyohara K, Kitamura T, Sakai T, Nishiyama C, Nishiuchi T, Hayashi Y, Sakamoto T, Marukawa S, Iwami T. Public- access AED pad application and outcomes for out-of-hospital cardiac arrests in Osaka, Japan. Resuscitation. 2016, ;106, pp.:70-75.
- State Laws on Cardiac Arrest and Defibrillators. NCSL. Available online: https://www.ncsl.org/research/health/laws-on-cardiac-arrest-and-defibrillators-aeds.aspx (accessed on 30.01.2020) r.].
- An Overview of State AED Laws and Recommendations. Available online: https://www.cardiopartners.com/blog/an-overview-of-state-aed-laws-and-recommendations (accessed on 10.02.2020 r.]).
- Cardiac Science. Seven-year AED study in Munich: Cardiac Science for the week of May 18. Available online. https://www.cardiacscience.com/seven-year-aed-study-in-munich-cardiac-science-for-the-week-of-may-18/ (accessed on: 10.02.2020 r).
- Installation of AED (Automated External Defibrillator) units. Osaka Metro. Available online: https://subway.osakametro.co.jp/en/guide/usage/aed/aed_0.php (accessed on: 10.02.2020 r)
- Gianotto‐Oliveira R, Gonzalez MM, Vianna CB, Alves MM, Timerman S, Filho RK, Karl B. Survival After Ventricular Fi-bril-lation Cardiac Arrest in the Sao Paulo Metropolitan Subway System: First Successful Targeted Automated External De-fib-rillator (AED) Program in Latin America. JAHA 2015, ;4(10).
- TOK FM. Hidden camera: "I can't give out the defibrillator!" Łukasz Wojtusik, TOK FM. Available online: https://wiadomosci.gazeta.pl/wiadomosci/1,114873,6265862,Ukryta_kamera___Nie_moge_wydac_defibrylatora__.html (ac-cessed on: 10.02.2020 r)
- Alqahtani SE, Alhajeri AS, Ahmed AA, Mashal SY. Characteristics of Out of Hospital Cardiac Arrest in the United Arab Emirates. Heart Views. 2019, ;20(4):, pp.146-151.
- Ringh M, Jonsson M, Nordberg P, Fredman D, Hasselqvist-Ax I, Håkansson F, Claesson A, Riva G, Hollenberg J. Survival after Public Access Defibrillation in Stockholm, Sweden A striking success. Resuscitation. 2015, ;91, pp.:1-7.
- Hansen CM, Wissenberg M, Weeke P, Ruwald MH, Lamberts M, Lippert FK, Gislason GH, Nielsen SL, Købe L, Torp-Pedersen C, Folke F. Automated External Defibrillators Inaccessible to More Than Half of Nearby Cardiac Arrests in Public Locations During Evening, Nighttime, and Weekends. Circulation 2013;, 128(20), pp.: 2224-2231.
- Bagai A, McNally B, Al-Khatib S, et al. Temporal differences in out-of-hospital cardiac arrest incidence and survival. Cir-cu-lation, 2013, ;128, pp.:2595-2602.
- Nadolny K, Gotlib J, Panczyk M, Ładny JR, Białczak Z, Podgórski M, Makar O, Izhytska N, Gałązkowski R. Epidemiology of sudden cardiac arrest in prehospital care in the Silesian Wiadomości Lekarskie 2018, ;71(1), pp.:193-200.
- Szczerbinski S, Ratajczak J, Lach P, et al. Epidemiology and chronobiology of out-of-hospital cardiac arrest in a subpopulation of southern Poland: A two-year observation. Cardiol J. 2018 [Epub ahead of print]
- Please transform “Anonymized data” (line 103) in a proper sentence.
Answer 4: Thank you very much for your valuable comment. We have corrected the sentence to:
The anonymized data included only the date and time of the event, age, gender, and treatment implemented.
- I recommend explaining the prepandemic context not only in the title but also in the abstract and manuscript.
Answer 5: Thank you very much for your valuable comment. This study will continue. The next analysis will include the years 2020-2022 and will be a comparative analysis to the current study.
- The article title is somewhat vague: I would reformulate it as follows: “Network of automated external defibrillators in Poland before the SARS-CoV-2 pandemic: an in-depth analysis”
Answer 6: Thank you very much for your valuable comment. We changed the topic to the proposed one: “Network of automated external defibrillators in Poland before the SARS-CoV-2 pandemic: an in-depth analysis”

Round 2
Reviewer 3 Report
Dear Sir/Madam,
I had the opportunity to act as a reviewer on the recent submission by Ślęzak et al. to the International Journal of Environmental Research and Public Health.
The authors have addressed almost all issues raised by the reviewers. However, the proofing of English language by a native speaker seems to be missing. For instance, on line 149, the authors write the following sentence “Presenting the place with the largest number of AEDs is Chopin Airport in Warsaw.” without using a predicate.
I recommend that the authors provide a substantially improved version of the manuscript before publishing, as this was an issue raised by all 3 reviewers.
Best regards
Author Response
Respected Reviewer thank you for your time, for your great contribution, we have corrected the work. I hope that now everything is correct.
Thank you, I wish you much health in this difficult time.